# Effects of replacement therapies with clotting factors in patients with hemophilia: A systematic review and meta-analysis

Carolina J. Delgado-Flores[1], David García-Gomero[2], Stefany Salvador-Salvador[1], José Montes-Alvis[3], Celina Herrera-Cunti[4], Alvaro Taype-Rondan[5]*

1 Instituto de Evaluación de Tecnologías en Salud e Investigación - IETSI, EsSalud, Lima, Peru, 2 Facultad de Medicina "San Fernando", Universidad Nacional Mayor de San Marcos, Lima, Peru, 3 EviSalud Evidencias en Salud, Lima, Peru, 4 Hospital Guillermo Almenara Irigoyen, EsSalud, Lima, Peru, 5 Unidad de Investigación para la Generación y Síntesis de Evidencias en Salud, Universidad San Ignacio de Loyola, Lima, Peru

* alvaro.taype.r@gmail.com

## Abstract

### Background

Different prophylactic and episodic clotting factor treatments are used in the management of hemophilia. A summarize of the evidence is needed inform decision-making.

### Objective

To compare the effects of factor replacement therapies in patients with hemophilia.

### Methods

We performed a systematic search in PubMed, Central Cochrane Library, and Scopus. We included randomized controlled trials (RCTs) published up to December 2020, which compared different factor replacement therapies in patients with hemophilia. Random-effects meta-analyses were performed whenever possible. The certainty of the evidence was assessed using the Grading of Recommendations Assessment, Development, and Evaluation (GRADE) methodology. The study protocol was registered in PROSPERO (CRD42021225857).

### Results

Nine RCTs were included in this review, of which six compared episodic with prophylactic treatment, all of them performed in patients with hemophilia A. Pooled results showed that, compared to the episodic treatment group, the annualized bleeding rate was lower in the low-dose prophylactic group (ratio of means [RM]: 0.27, 95% CI: 0.17 to 0.43), intermediate-dose prophylactic group (RM: 0.15, 95% CI: 0.07 to 0.36), and high-dose prophylactic group (RM: 0.07, 95% CI: 0.04 to 0.13). With significant difference between these subgroups (p = 0.003, I² = 82.9%). In addition, compared to the episodic treatment group, the annualized joint bleeding rate was lower in the low-dose prophylactic group (RM: 0.17, 95% CI: 0.06 to

**Funding:** The author(s) received no specific funding for this work.

**Competing interests:** The authors have declared that no competing interests exist.

0.43), intermediate-dose prophylactic group (RM of 0.14, 95% CI: 0.07 to 0.27), and high-dose prophylactic group (RM of 0.08, 95% CI: 0.04 to 0.16). Without significant subgroup differences. The certainty of the evidence was very low for all outcomes according to GRADE methodology. The other studies compared different types of clotting factor concentrates (CFCs), assessed pharmacokinetic prophylaxis, or compared different frequencies of medication administration.

## Conclusions

Our results suggest that prophylactic treatment (at either low, intermediate, or high doses) is superior to episodic treatment for bleeding prevention. In patients with hemophilia A, the bleeding rate seems to have a dose-response effect. However, no study compared different doses of prophylactic treatment, and all results had a very low certainty of the evidence. Thus, future studies are needed to confirm these results and inform decision making.

## Introduction

Hemophilia refers to an X-linked bleeding disorder caused by a deficiency of coagulation factor VIII (hemophilia A) or factor IX (hemophilia B) [1]. It affects more than 1.2 million individuals worldwide in 2017 [2]. Hemophilia causes bleeding, specially hemarthrosis, which represents up to 80 percent of hemorrhages, is painful, can be physically debilitating, and may lead to permanent disability [3]. To prevent bleeding in these patients, prophylactic and episodic therapies with factor administration are widely used, which are effective but expensive treatments.

Prophylactic therapy, defined as factor administration in the absence of bleeding, is a therapeutic strategy to reduce bleeding and its long-term complications such as chronic arthropathy, especially in severe hemophilia (factor VIII or IX activity <1% of normal) [4]. On the other hand, episodic or on-demand therapy is referred the factor administration in the presence of bleeding and has been proposed as an alternative in the context of mild or moderate factor deficiency with a decreased clinical bleeding phenotype [5].

The World Federation of Hemophilia in 2020, referred that prophylactic therapy is preferred in comparison to episodic therapy in children and, if possible, should be continued in adulthood. In that context, they suggest the administration of factors VIII or IX with standard half-life clotting factor at high or intermediate doses [6]. However, in lower or -middle-income countries like India, the local consensus suggests the use of a low-dose of prophylactic therapy [7], while guidelines of other countries such as Chile in 2013 [8], Argentina in 2015 [9], Peru in 2016 [10], and Colombia in 2015 [11] recommend the on-demand therapy but do not consider the low-dose prophylaxis like an option.

Considering that the costs and burdens of prophylaxis are high, and the recommendations of the available guidelines are mainly based on expert consensus or systematic reviews with serious limitations, we performed a systematic review that aims to assess the effects of factor replacement therapies in patients with hemophilia.

## Material and methods

### Protocol and registration

We performed a systematic review and meta-analysis following the Preferred Reporting Items for Systematic Reviews and Meta-analysis (PRISMA) recommendations [12]. The study protocol has been registered at PROSPERO (CRD42021225857).

## Information sources, search and study selection

For this systematic review, we included all randomized controlled trials (RCTs) that compared the effects of different factor replacement therapies (such as prophylactic, episodic, tailored, or other therapies) in patients with hemophilia. We only included those studies that were published at length in scientific journals.

Searching was performed in two steps: 1) a systematic search in three databases, and 2) a review of all the references cited in any of the studies included in step 1. Both steps were performed independently by two reviewers (CJDF and DGG). When disagreements occurred, they were discussed by all authors and resolved by consensus.

To carry out step 1, we performed a literature search in the following databases and search engines: PubMed, Cochrane Central Register of Controlled Trials (CENTRAL), and Scopus. No restrictions in language or publication date were applied. The last research update was performed in December 2020. The detailed search strategy for this step is available on **Table in** S1 Table. We downloaded all found references to an EndNote document, and eliminated duplicated references using this software. After that, we assessed titles and abstracts to identify potential studies for inclusion. Lastly, we assessed the full-text of these potential studies to determine their eligibility.

For step 2, during December 2020, we reviewed all the references of the studies included in step 1. Later, we collected all articles that met the inclusion criteria.

## Data extraction

Two independent authors (CJDF and DGG) extracted the following information of the included studies into a Microsoft Excel worksheet: author, year of publication, countries or regions, population (hemophilia type, age and sex), factor activity level, product (types of clotting factor concentrates [CFCs] and half-life in hours) [13], control (sample, dose, and frequency), intervention (sample, dose, and frequency), follow-up, and funding. When disagreements were found, the full-text articles were reviewed again by the authors.

The factor replacement therapies for hemophilia A were categorized according to the weekly doses used, based in the World Federation of Hemophilia Guidelines 2020 [6] as following: low-dose (20 to <45 IU/kg per week), intermediate-dose (45 to <75 IU/kg per week), high-dose (≥75 IU/kg per week), and pharmacokinetic [PK]-prophylaxis (which adjust the prophylaxis dose and frequency after pharmacokinetic evaluations of each patient).

## Risk of bias and certainty of the evidence

To evaluate the risk of bias of included RCTs, we used The Cochrane Collaboration's tool for assessing risk of bias [14]. This tool assesses the risk of bias in seven domains per outcome of interest: random sequence generation, allocation concealment, blinding of participants and personnel, blinding of outcome assessment, incomplete outcome data, selective reporting, and other sources of bias. For each of the domains, the overall risk of bias (low risk, unclear risk, and high risk) was established according to the judgment of their *signaling* questions.

To assess the certainty of the evidence for each outcome, we used The Grading of Recommendations Assessment, Development and Evaluation (GRADE) methodology [15], which evaluates the study design, risk of bias, inconsistency, indirectness, imprecision, and publication bias.

## Statistical analysis

For count outcomes such as the number of bleeding episodes, we calculated and reported the intervention effects as ratio of means (RM), defined as mean of the outcome in the intervention group / mean of the outcome in the control group. For dichotomous outcomes, we used risk ratios (RR). In all cases, we showed the 95% confidence intervals (95% CIs). For studies in which the standard deviations (SD) were missing, we imputed them using linear regressions taking into account the outcome means and SDs of the other included studies.

When two or more studies presented the same outcome in a similar fashion, we performed a meta-analysis using random-effects models (Inverse Variance and Mantel-Haenszel method) due to heterogeneity across studies interventions [16]. Meta-analyses were performed using the software Review Manager 5.4.1.

We assessed heterogeneity using the $I^2$ statistics, and we considered that heterogeneity might not be important when $I^2 < 40\%$ [14]. Publication bias was not assessed due to the number of studies pooled for each meta-analysis were less than ten [14].

# Results

## Studies selection

We found 1563 records in databases searching. After duplicates removal, we screened 1085 records, from which we reviewed 93 full-text documents, and finally included 11 documents from 9 studies. The complete list of articles that were excluded in the full-text assessment is detailed in **Table in** S2 Table. Then, we searched the references of the included studies. However, no extra articles that fulfilled our inclusion criteria were found in these searches (Fig 1).

The 11 included documents reported results of 9 RCTs. Two papers reported results from the Joint Outcome Study: Hacker 2007 [17] and Manco-Johnson 2007 [18]; and other two papers reported results from SPINART study: Manco-Johnson 2013 [19] and Manco-Johnson 2017 [20]. We will cite the papers by Manco-Johnson 2007 and Manco-Johnson 2017 to refer to each study, since both were the main papers of their respective studies.

## Characteristics

Of the included 9 RCTs, six compared episodic vs prophylactic treatments [18, 20–24], while the other three performed other comparisons [25–27]. Of the 9 studies, five [20, 22, 25–27] were multicenter, conducted in different countries in Europe, South Africa, North America, South America, and Asia; and the other four studies [18, 21, 23, 24] were conducted in a single country: The United States, Italy, India, and Indonesia. Sample size ranged from 21 to 131 patients. Regarding the population characteristics, eight studies [18, 20–24, 26, 27] were performed in patients with hemophilia A, and one study [25] in patients with hemophilia B. Four studies [18, 21, 23, 24] included children, while the other five studies [20, 22, 25–27] included children and adults. Six studies [20–24, 27] included severe hemophilia (< 1% factor activity level), and three studies [18, 25, 26] included moderately severe or severe hemophilia ($\leq$ 2% factor activity level) (Table 1).

## Risk of bias

Overall, in most of the studies, the items of the Cochrane tool were rated as high or unclear risk of bias. Mainly for the allocation concealment (7/9 studies had an unclear risk of bias), blinding of participants and personnel (8/9 studies had a high risk of bias), blinding of outcome assessment (8/9 studies had an unclear risk of bias), and incomplete outcome data (6/9 studies had a high risk of bias) (Fig 2).

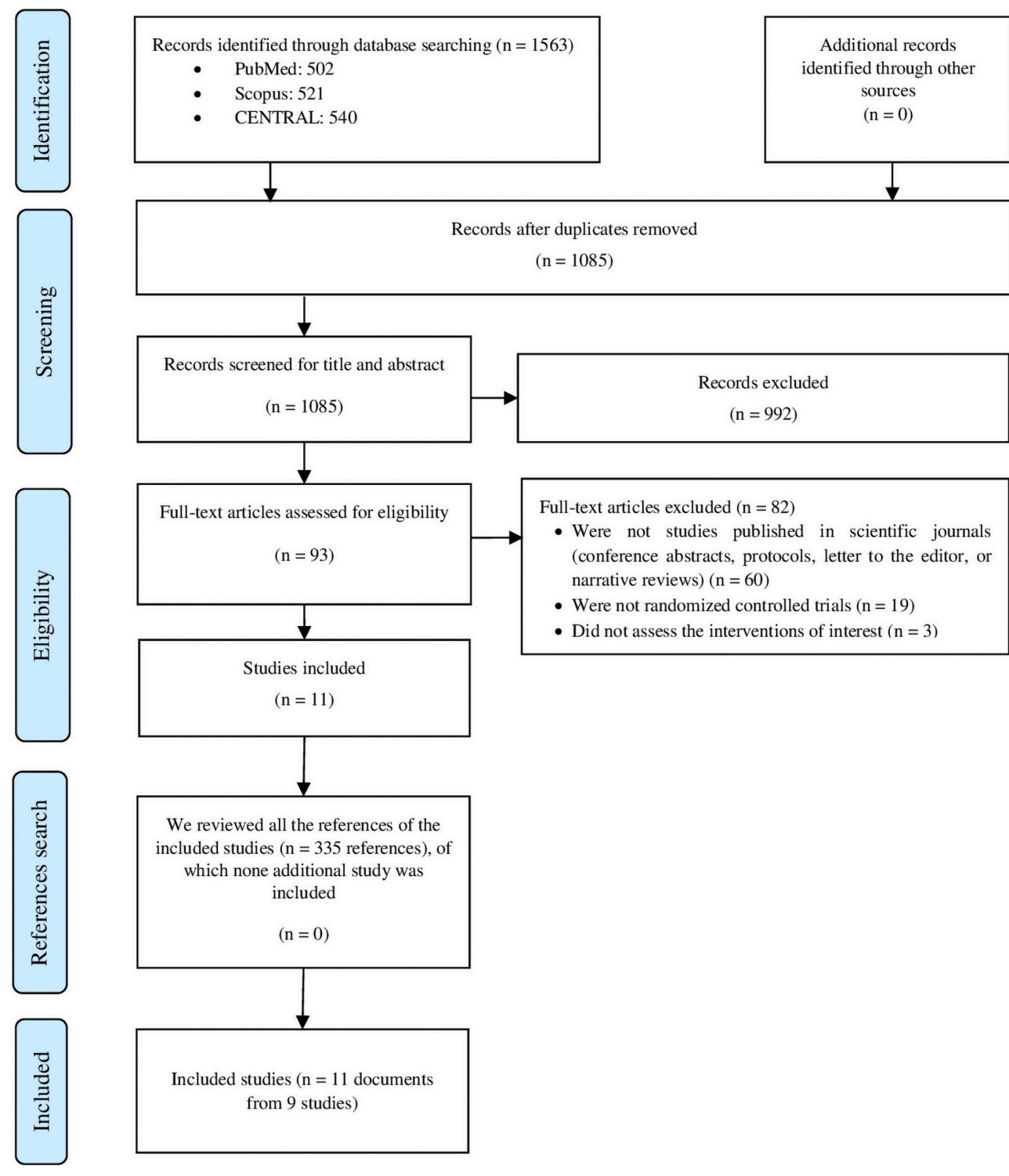

**Fig 1. Flow diagram (study selection).**

**Table 1. Study and participants' characteristics in the included RCTs.**

| N | Author (year) | Countries or regions | Population: hemophilia type, age and sex | Factor activity level** | Product: type of clotting factor concentrates and half-life (hours) | Control (n) | Intervention (n) | Follow-up | Fun-ding |
|---|---|---|---|---|---|---|---|---|---|
| colspan="10" | **Episodic treatment compared with prophylaxis (at low, intermediate, and high doses)** | | | | | | | | |
| 1 | Verma (2016) | India | • Hemophilia A<br>• Age range: 1 to 10 yr (mean: 6.11 yr)<br>• Sex: not mentioned | < 1% | **FVIII concentrate (Hemofil M)**<br>• Plasma-derived, mAb-purified<br>• 15 h | **Episodic (n = 10)**<br>1. 25 IU/kg or more as early as possible after the joint bleed,<br>2. 25 IU/kg every 12–24 h until resolution | **Low-dose prophylaxis (n = 11)**<br>• **Weekly dose: 20 IU/kg** (10 IU/kg twice a week) | Median: 0.96 yr | Self-funded |
| 2 | Chozie (2019) | Indonesia | • Hemophilia A<br>• Age range: 4 to 18 yr (mean: 11.95 yr)<br>• Sex: not mentioned | < 1% | **FVIII concentrate (Koate-DVI)**<br>• Plasma-derived, chromatography purified<br>• 16 h | **Episodic (n = 25)**<br>• Not specified | **Low-dose prophylaxis (n = 25)**<br>• **Weekly dose: 20 IU/kg** (10 IU/kg twice a week) | Mean: 1 yr | Grifols |
| 3 | Manco-Johnson (2007) and Hacker (2007) | United States | • Hemophilia A<br>• Age range: 1 to 2.5 yr (mean: 1.6 yr)<br>• Sex: 100% males | ≤ 2% | **FVIII concentrate (Kogenate or Kogenate FS)**<br>• Recombinant<br>• 11 to 15 h | **Episodic (n = 33)**<br>1. 40 IU/kg at the time of joint hemorrhage.<br>2. 20 IU at 24 hours and 72 hours after the first dose<br>3. 20 IU/kg every second day, until 4 weeks. | **Intermediate-dose prophylaxis (n = 32)**<br>• **Weekly dose: 75 IU/kg** (25 IU/kg every second day) | Mean: 4.08 yr | CDC, NIH, Bayer |
| 4 | Gringeri (2011) | Italy | • Hemophilia A<br>• Age range: 1 to 7 yr (mean: 4.10 yr)<br>• Sex: not mentioned | < 1% | **FVIII concentrate (Recombinate® until 2003 / Advate® since 2004)**<br>• Both were recombinant<br>• Recombinate: 15 h / Advate: 9 to 12 h<br>• 1° generation / 3° generation | **Episodic (n = 19)**<br>1. 25 IU/kg or more, possibly within 6 h from the bleeding,<br>2. Repeated every 12–24 h until complete resolution | **Intermediate-dose prophylaxis (n = 21)**<br>• **Weekly dose: 75 IU/kg** (25 IU/kg three times a week) | Median: 6.88 yr | Baxter |
| 5 | Manco-Johnson (2014) and Manco-Johnson (2017) | United States, Bulgaria, Romania and Argentina | • Hemophilia A<br>• Age range: 12 to 50 yr (mean: 29 yr)<br>• Sex: 100% males | < 1% | **FVIII concentrate (Kogenate FS)**<br>• Recombinant<br>• 11 to 15 h | **Episodic (n = 42)**<br>• Not specified | **Intermediate-dose prophylaxis (n = 41)**<br>• **Weekly dose: 75 IU/kg** (25 IU/kg three times a week) | 3 yr | Bayer |
| 6 | Kavakli (2015) | Europe, South Africa, North America, South America, and Asia | • Hemophilia A<br>• Age range: 12 to 65 yr (mean: 29.6 yr)<br>• Sex: 100% males | < 1% | **FVIII concentrate (BAY 81–8973, Kovaltry)**<br>• Recombinant<br>• 12 to 14 h | **Episodic (n = 21)**<br>• Dependent on the location and severity of the bleed | **Intermediate-dose prophylaxis (n = 28)**<br>• **Weekly dose: 40 to 60 IU/kg** (20–30 IU/kg twice a week)<br><br>**High-dose prophylaxis (n = 31)**<br>• **Weekly dose: 90 to 120 IU/kg** (30–40 IU/kg three times a week) | 1 yr | Bayer |

*(Continued)*

**Table 1.** (Continued)

| N | Author (year) | Countries or regions | Population: hemophilia type, age and sex | Factor activity level** | Product: type of clotting factor concentrates and half-life (hours) | Control (n) | Intervention (n) | Follow-up | Fun-ding |
|---|---|---|---|---|---|---|---|---|---|
| **Studies that compared two different prophylactic factors** | | | | | | | | | |
| 1 | Powell (2012) | United States, Israel, Poland, Italy, Austria, and Denmark | • Hemophilia A<br>• Age range: 12 to 70 yr (mean: 33.6 yr)<br>• Sex: 100% males | < 1% | **Intervention group** Kogenate FS reconstituted with a pegylated liposome solvent (BAY 79–4980)<br>• Recombinant<br>• 11 to 15 h<br><br>**Control group** rFVIII-FS (Kogenate FS)<br>• Recombinant<br>• 11 to 15 h | **rFVIII-FS (n = 68)**<br>• **Weekly dose: 75 IU/kg** (25 IU/kg three times a week) | **BAY 79–4980 (n = 63)**<br>• **Weekly dose: 35 IU/kg** (35 IU/kg once a week) | Median: 0.96 yr | Bayer |
| **Studies that assessed the pharmacokinetic prophylaxis** | | | | | | | | | |
| 1 | Valentino (2012) | United States and Europe | • Hemophilia A<br>• Age range: 7 to 65 yr (median: 27.5 yr)<br>• Sex: 100% males | ≤ 2% | **FVIII concentrate (Advate)**<br>• Recombinant<br>• 9 to 12 h | **Intermediate- to high-dose prophylaxis (n = 32)**<br>• **Weekly dose: 70 to 140 IU/kg** (20–40 IU/kg every 48 ± 6 h) | **PK-prophylaxis (n = 34)**<br>• **Weekly dose: 46.7 to 186.7 IU/kg** (20–80 IU/kg every 72 ±6 h) | Mean: 0.96 yr | Baxter |
| **Studies in which the groups received the same weekly doses but with different frequency** | | | | | | | | | |
| 1 | Valentino (2014) | United States, Canada, and Europe | • Hemophilia B<br>• Age range: 6 to 65 yr (mean: 28.4 yr)<br>• Sex: 100% males | ≤ 2% | **FIX concentrate (BeneFIX)**<br>• Recombinant<br>• 16 to 19 h | **High-dose prophylaxis (n = 22)**<br>• **Weekly dose: 100 IU/kg** (100 IU/kg once a week) | **High-dose prophylaxis (n = 25)**<br>• **Weekly dose: 100 IU/kg** (50 IU/kg twice a week) | 32 weeks (0.62 yr) | Pfizer |

**yr**: years; **IQR**: Interquartile range; **SD**: Standard deviation; **mAb-purified**: monoclonal antibody-purified; **rFVIII-FS**: Sucrose-formulated rFVIII.

*All studies excluded patients with inhibitors.

**Factor VIII for all studies performed in patients with haemophilia A, or factor IX for the study performed in patients with haemophilia B.

## Episodic vs prophylactic treatment

First, we will focus on the six RCTs that compared episodic vs prophylactic treatments (at either low, intermediate, or high doses) [18, 20–24]. These studies reported several outcomes, such as annualized bleeding rate (ABR), annualized joint bleeding rate (AJBR), radiographic findings (which were meta-analyzed and reported in Fig 3 and Table 2), hemophilia joint health score 2.1 (HJHS-2.1), joint structural changes (using extended magnetic resonance imaging-eMRI), Petterson score, adverse events (AEs), quality of life (reported in Table 2), and other secondary outcomes (detailed in **Table in** S3 Table).

ABR was assessed in six studies 18, 20–24], which follow-up ranged from 0.96 to 6.88 years. We performed meta-analyses by sub-groups according to the dose used in the prophylactic treatment. These analyses showed that, compared to the group that used episodic treatment, mean ABR was lower in those who used a low-dose prophylaxis (RM: 0.27, 95% CI: 0.17 to 0.43), intermediate-dose prophylaxis (RM: 0.15, 95% CI: 0.07 to 0.36), and high-dose prophylaxis (RM: 0.07, 95% CI: 0.04 to 0.13). With significant difference between these subgroups

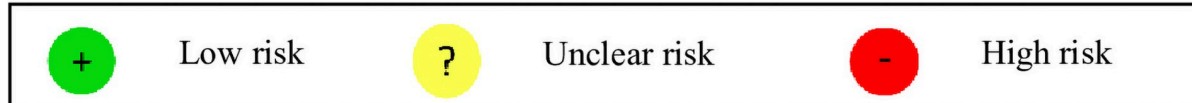

**Fig 2. Risk of bias of the included studies.**

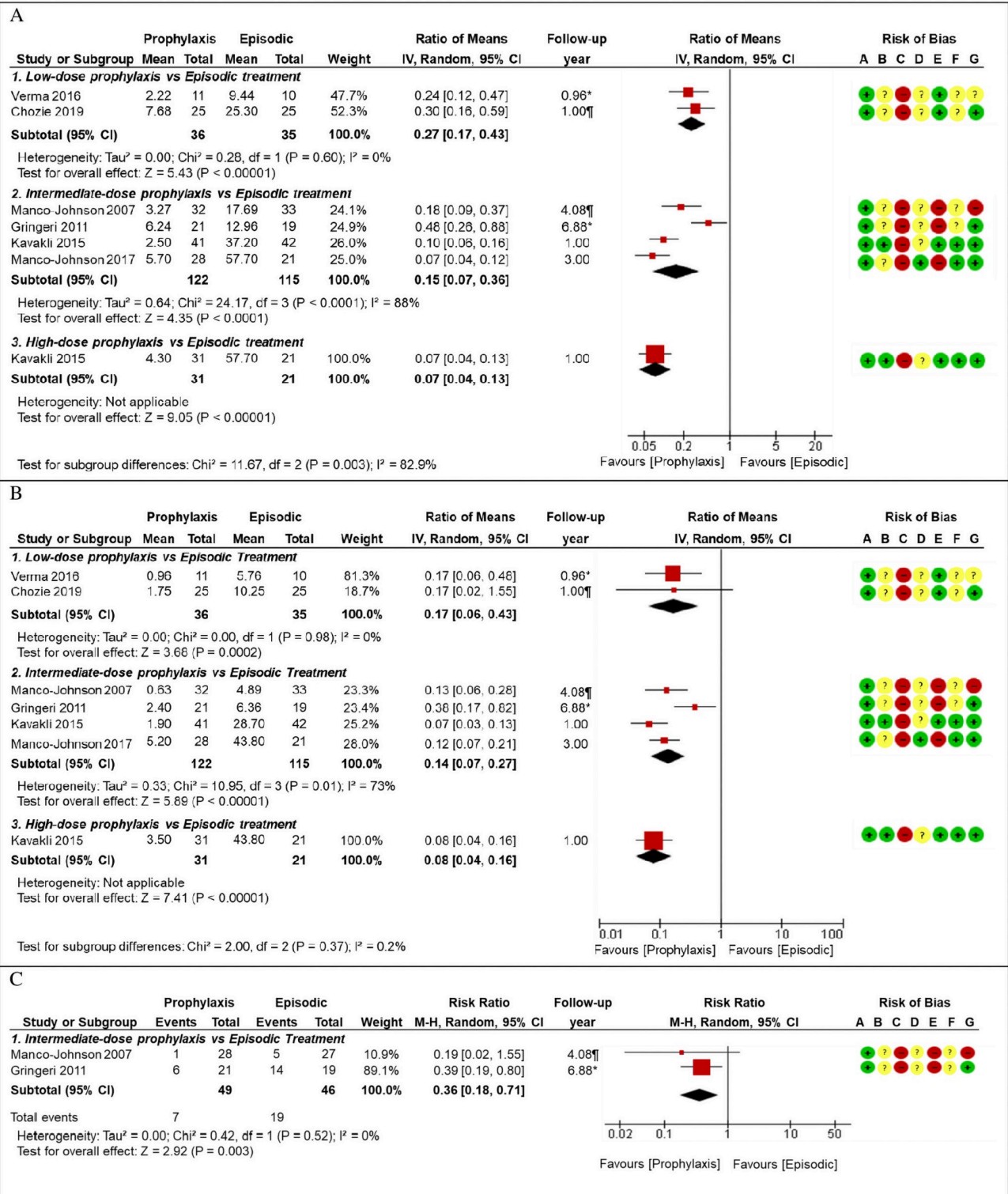

**Fig 3. Forest plot for episodic versus prophylactic factor replacement therapies.** *Mean, Median. (A) Forest plot for annualized bleeding rate, episodic treatment as control group. (B) Forest plot for annualized joint bleeding rate, episodic treatment as control group. (C) Forest plot for radiographic findings, episodic treatment as control group.

**Table 2.  Summary of findings for episodic treatment vs prophylaxis (either low, intermediate, or high dose).**

| Outcomes (follow-up in months) | of participants (studies) | Anticipated absolute effects* (95% CI) | | Relative effect (95% CI) | Certainty of the evidence (GRADE) |
|---|---|---|---|---|---|
| | | Risk with Control | Risk with Intervention | | |
| **Episodic treatment (control) vs Low-dose prophylaxis (intervention)** | | | | | |
| Annualized bleeding rate (12 m) | 71 (2 RCTs) | Range of means: 9.4–25.3 | Range of means: 2.2–7.7 | RM 0.27 (0.17 to 0.43) | ⊕◯◯◯ VERY LOW [a,d] |
| Annualized joint bleeding rate (12 m) | 71 (2 RCTs) | Range of means: 5.8–10.3 | Range of means: 1.0–1.8 | RM 0.17 (0.06 to 0.43) | ⊕◯◯◯ VERY LOW [a,d] |
| Change in the Hemophilia joint health score-2.1 (HJHS-2.1). Range: 0 to 124. Higher score = worst (12 m) | 66 (2 RCTs) | • **Verma 2016** Low-dose prophylaxis: median change of 0 points. Episodic treatment: median change of 4.5 points (p<0.05). • **Chozie 2019** Low-dose prophylaxis: median change of -1 points. Episodic treatment: median change of 2 points (p<0.001). | | | ⊕◯◯◯ VERY LOW [a,d] |
| Change in the Petterson score (11.5 m) | 21 (1 RCT) | • **Verma 2016** Low-dose prophylaxis: median change of 0 points. Episodic treatment: median change of 1 point (no p-value provided). | | | ⊕◯◯◯ VERY LOW [a,d,e] |
| **Episodic treatment (control) vs Intermediate-dose prophylaxis (intervention)** | | | | | |
| Annualized bleeding rate (12.0 to 82.5 m) | 237 (4 RCTs) | Range of means: 13.0–57.7 | Range of means: 2.5–6.2 | RM 0.15 (0.07 to 0.36) | ⊕◯◯◯ VERY LOW [a,b,c] |
| Annualized joint bleeding rate (12.0 to 82.5 m) | 237 (4 RCTs) | Range of means: 4.9–43.8 | Range of means: 0.6–5.2 | RM 0.14 (0.07 to 0.27) | ⊕◯◯◯ VERY LOW [a,b,c] |
| Radiographic findings (49.0 to 82.5 m) | 95 (2 RCTs) | 413 per 1000 | 149 per 1000 | RR 0.36 (0.18 to 0.71) | ⊕◯◯◯ VERY LOW [a,d] |
| Joint structural changes (using eMRI scores). Range: 0 to 45. Higher score = worst (36 m) | 83 (1 RCT) | • **Manco-Johnson 2017:** Intermediate-dose prophylaxis: mean change of 0.79 points. Episodic treatment: mean change of 0.96 points (p = 0.66). | | | ⊕◯◯◯ VERY LOW [a,d,e] |
| Petterson score at the end of the follow-up (82.5 m) | 40 (1 RCT) | • **Gringeri 2011:** Intermediate-dose prophylaxis group (n = 6, median Pettersson score of 5). Episodic treatment group (n = 14, median Pettersson score of 8), p<0.05. | | | ⊕◯◯◯ VERY LOW [a,d,e] |
| Quality of life (36.0 to 82.5 m) | 123 (2 RCTs) | • **Gringeri 2011 (82.5 months):** Score in the "family" dimension of the Haemo-QoL scale was lower (better) in patients with intermediate-dose prophylaxis (mean: 11.3) than in those with episodic treatment (mean 44.0), p<0.05. • **Manco-Johnson 2017 (36 months):** • Mean change in the score of the Haemo-QoL-A: Intermediate-dose prophylaxis group: 3.98 points. Episodic treatment: 6.00 points (p = 0.27). • Mean change in the score of the EQ VAS (higher = better): Intermediate-dose prophylaxis: 10.49 points. Episodic treatment: –1.80 points. No p-value provided. • Mean change in the EQ-5D utility index score (higher = better): Intermediate-dose prophylaxis: 0.06 points. Episodic treatment: –0.01 points. No p-value provided. | | | ⊕◯◯◯ VERY LOW [a,d] |

*(Continued)*

**Table 2.** (Continued)

| Outcomes (follow-up in months) | of participants (studies) | Anticipated absolute effects* (95% CI) | | Relative effect (95% CI) | Certainty of the evidence (GRADE) |
|---|---|---|---|---|---|
| | | Risk with Control | Risk with Intervention | | |
| Adverse events (12.0 to 82.5 m) | 154 (3 RCTs) | • **Gringeri 2011:**<br>  • Inhibitors developing: 3/21 patients in the prophylaxis group and 2/19 in the episodic group.<br>  • CVAD-related infection: 6/20 patient in the prophylaxis group, and 0/19 in the episodic group (no indwelling catheters required).<br>• **Manco-Johnson 2007** reported that 6/32 patients had CVAD-related infection in the prophylaxis group and 6/33 in the episodic group. | | | ⊕◯◯◯<br>VERY LOW [a,d] |
| **Episodic treatment (control) vs High-dose prophylaxis (intervention)** | | | | | |
| Annualized bleeding rate (12 m) | 52 (1 RCT) | Mean: 57.7 | Mean: 4.3 | RM 0.07 (0.04 to 0.13) | ⊕◯◯◯<br>VERY LOW [a,d] |
| Annualized joint bleeding rate (12 m) | 52 (1 RCT) | Mean: 43.8 | Mean: 3.5 | RM 0.08 (0.04 to 0.16) | ⊕◯◯◯<br>VERY LOW [a,d] |

*The risk in the intervention group (and its 95% confidence interval) is based on the assumed risk in the comparison group and the relative effect of the intervention (and its 95% CI).

**CI**: Confidence interval; **yr**: years **RM**: ratio of means; **RR**: Risk ratio; **Haemo-QoL**: Hemophilia quality of life questionnaire for children; **Haemo-QoL-A**: Hemophilia-specific quality of life questionnaire for adults; **EQ VAS**: EuroQol visual analogue scale; **SD**: Standard deviation; **CVAD**: Central venous access device-related infections.

**Explanations**

[a]. We rated down one level for risk of bias.

[b]. We rated down one level for imprecision due to the small number of participants that presented the outcome (200–400).

[c]. We rated down one level for inconsistency ($I^2 > 70\%$).

[d]. We rated down two levels for imprecision due to the small number of participants that presented the outcome (less than 200)

[e]. We rated down one level for publication bias.

(test for subgroup differences: p = 0.003, $I^2$ = 82.9%) (Fig 3A). These results had a very low certainty of evidence (Table 2).

AJBR was assessed in six studies [18, 20–24], which follow-up ranged from 0.96 to 6.88 years. We performed meta-analyses by sub-groups according to the dose used in the prophylactic treatment. These analyses showed that, compared to those that used episodic treatment, AJBR was lower in those who used a low-dose prophylaxis (RM: 0.17, 95% CI: 0.06 to 0.43), intermediate-dose prophylaxis (RM: 0.14, 95% CI: 0.07 to 0.27), and high-dose prophylaxis (RM: 0.08, 95% CI: 0.04 to 0.16). Without finding significant difference between these sub-groups (test for subgroup differences: p = 0.37, $I^2$ = 0.2%) (Fig 3B). This result had a very low certainty of evidence (Table 2).

The presence of radiographic findings was assessed in two studies [18, 24], which follow-up ranged from 4.08 to 6.88 years. The meta-analysis showed that, compared to those that used episodic treatment, those who received intermediate-dose prophylaxis had a lower rate of having radiographic findings (RR: 0.36, 95% CI: 0.18 to 0.71) (Fig 3C). This result had a very low certainty of evidence (Table 2).

HJHS-2.1 was assessed in two studies [21, 23], which compared episodic treatment versus low-dose prophylaxis, finding that the median score was lower in the prophylaxis group (which means a beneficious effect). Regarding quality of life, Gringeri 2011 found that the group that received intermediate-dose prophylaxis had a better quality of life in the "family"

dimension of the Haemo-QoL, compared to those who were in the episodic treatment group. These results had a very low certainty of evidence (Table 2).

AEs were reported only for the studies that compared intermediate-dose prophylaxis versus episodic treatment. These studies reported the developing of inhibitors (prophylaxis group: 3/21; episodic: 2/19) [24], and CVAD-related infection (prophylaxis group: 6/20 and 6/32; episodic: 0/19 and 6/33) [18, 24]. This result had a very low certainty of evidence (Table 2).

Additionally, we found other outcomes that were detailed in the Summary of Findings (**Table in** S3 Table): Joint physical examination (using Colorado adult joint assessment scale-CAJAS), pain (short-form McGill pain questionnaire), quality of life (HRQoL), change in activity level, healthcare resource utilization, treatment satisfaction, and hemophilia early arthropathy detection with ultrasound (HEAD-US), adverse events.

## Other comparisons

Powell 2012 [27] compared two different pharmaceutical products (intervention group: BAY 79–4980 at 35 IU/kg once a week, control group: rFVIII-FS at 25 IU/kg three times a week), finding that ABR and AJBR were higher in patients that used BAY 79–4980 than in those with rFVIII-FS (very low certainty of the evidence) (**Table in** S3 Table).

Valentino 2012 (n = 66) [26] compared intermediate-to-high-dose prophylaxis (70 to 140 IU/kg weekly) with pharmacokinetic prophylaxis (46.7 to 186.7 IU/kg weekly). The pharmacokinetic prophylaxis dose was adjusted using the following formula $D = (2^{72/t})/r$, where **D**: dose, **72**: infusion interval [hours], **t**: terminal half-life (time required to decrease plasma concentration by 50%) [hours], and **r**: incremental recovery (peak factor level recorded in the first hour after infusion) [IU/mL]/[IU/kg]. This formula was based in two studies that proposed models factor replacement therapies adjustment according to the pharmacokinetic profile [28, 29]. This study reported no statistical differences in ABR and AEs outcomes between PK-prophylaxis and intermediate-dose prophylaxis (very low certainty of the evidence) (**Table in** S3 Table).

Valentino 2014 [25] administrated the same doses of Nonacog alfa at different time intervals (50 IU/kg twice a week, vs 100 IU/kg once a week). It reported mean ABR of 2.6 in the twice-a-week group and 4.6 in the weekly group (p = 0.217), and mean AJBR of 1.9 in the twice-a-week group and 3.6 in the weekly group (no p-value provided). These results had a very low certainty of the evidence. (**Table in** S3 Table).

## Discussion

Our results suggest that prophylaxis treatments have a higher benefit in comparison to an episodic treatment, and a dose-response effect (higher prophylaxis dose related to higher benefit) was observed for ABR but not for AJBR. While this suggest that the doses have a greater impact in ABR than in AJBR, the small sample sizes and the lower number of joint bleedings compared with total bleedings could have hindered the dose-response effect for AJBR.

These results, however, need to be taken with caution, since they had a low certainty of the evidence, and only one study was included for the comparison between episodic treatment and high-dose prophylaxis (n = 52), two studies for the comparison between episodic treatment and low-dose prophylaxis (n = 71), and four studies for the comparison between episodic treatment and intermediate-dose prophylaxis (n = 237). Moreover, studies were performed across different countries with different health systems, and using different types of CFCs; and we did not find any RCT that has compared different doses of prophylactic treatment, which are required to have accurate estimates regarding the impact of different prophylaxis doses.

Accordingly, previous sequential-treatment studies in children and adults with hemophilia A or B found that patients using intermediate- and low-doses prophylaxis had lower ABR and lower AJBR than those with episodic treatment [30–32].

Other non-randomized studies have compared different CFC doses. Two prospective observational studies showed lower ABR and lower AJBR in patients with intermediate- dose, than those with low-dose; although both studies did not adjust for confounding factors [33, 34].

Also, two studies compared high- versus intermediate- CFC doses. One observational study showed lower AJBR and better joint health (HJHS) in patients with high-dose, than those with intermediate-dose; did not adjust for confounding factors [35]. Other observational study reported lower AJBR in patients with high-dose, compared with intermediate-dose, after adjustment for age [36].

For ABR, we found a high heterogeneity in the meta-analysis that compared episodic treatment with intermediate-dose prophylaxis. This heterogeneity may be explained by the differences in the number of annualized bleedings of the control (episodic) group, as follows: two of the four meta-analyzed studies had a lower mean ABR in their episodic groups (17.69 and 12.96), while the other two studies had a higher mean ABR in their episodic groups (37.20 and 57.70). Thus, although the mean ABR was low in the prophylaxis group of the four studies (range: 2.50 to 6.24), the RM showed a lower benefit in those studies with lower number of bleedings in the control group, in which it would have been necessary to achieve a mean ABR very close to zero in their prophylaxis group to find an effect similar to that of the other two studies. Also, it is important to note that the two studies that had a higher mean ABR in their episodic group were the only ones that included adults (with a mean of 29.0 and 29.6 years) and were multicenter studies, while the other two had a lower age (with a mean of 1.6 and 4.1 years) and were carried out in a single country. A similar heterogeneous result was found in the meta-analysis that compared AJBR between the episodic and the intermediate-dose prophylaxis treatments, which included the same studies than the meta-analysis performed for ABR.

CFCs had different characteristics across studies, those included in la meta-analysis used plasma-derived (2/6) and recombinant concentrates (4/6); and all had a standard half-life. Currently, according to the World Federation of Hemophilia Guidelines 2020, both types of CFCs (plasma-derived and recombinant) are the treatment of choice for hemophilia, since both of them are considered as safe and effective for treating and preventing bleeds [6]. However, two previous studies (ECA and cohort) that have assessed the risk of inhibitor development between plasma-derived and recombinant CFCs showed discrepancies between their results [37, 38], so it is necessary assess that future RCTs compare between different types of CFCs.

We only have found one RCT that have compared different CFCs (BAY 79–4980 vs rFVIII-FS, both recombinant), suggesting that rFVIII-FS had lower bleeding. Also, one RCT that have compared PK-prophylaxis vs fixed dose prophylaxis, and one compared different dosing intervals. Both studies did not find significant differences, so future well-designed studies are needed.

Prophylaxis is an expensive treatment which requires that significant resources are allocated to hemophilia care, which poses a definite barrier to patient access [39]. Thus, health systems may choose to perform economic analyses in order to decide which therapy and which doses will recommend for the treatment of their hemophilia patients. These analyses should take into account the available budget, the costs, and the possible savings in terms of acute and chronic consequences of the bleeding [6, 39, 40].

A systematic review of cost-utility for hemophilia included 11 studies published from 2000 to 2015 (mostly from United Kingdom and the United States), which used Markov models

with 3-months to 1-year cycle length [40]. This review found that, compared with episodic treatment, prophylactic treatment had a median Incremental Cost-Effectiveness Ratios (ICER) of $86,000 per QALY gained for severe hemophilia A and $17,000 per QALY for hemophilia B. However, this study do not calculate the ICER for different scenarios using different doses (low, intermediate, high, or tailored doses) [40].

All studies found in our systematic review assessed interventions that used CFCs with standard half-life. CFCs with extended half-life require fewer CFCs administration per week due to their pharmacokinetic properties, which may improve the quality of life of the patients; and also require less use of medical devices, which could help lower the cost of prophylactic treatment. Thus, there is a need of RCTs assessing the use of extended half-life CFCs, which are believed to be as safe and effective as standard half-life CFCs, as WFH 2020 suggests [6].

## Strengths and limitations

This is an up-to-date summarize of the RCTs that have assessed different types of replacement therapies with CFCs in patients with hemophilia, which provides important information for decision-making in this regard. Our search was performed in three databases, which we believe contain the most important scientific contributions around the world [41, 42]. Also, we searched in the reference lists of the included studies. Thus, we are confident that all relevant literature RCTs are included.

However, the body of evidence shows significant limitations: 1) studies had a heterogeneous follow-up period (between eight months and 82.5 months), and used different types of replacement therapies; which difficult the comparability of their results. 2) All of the studies that compared high- or low-dose of prophylaxis with episodic treatment were performed in children, and all studies were performed in patients with hemophilia A. Thus, extrapolation to adults and patients with hemophilia B should be made with caution. 3) ABR and AJBR rate outcomes were measured as self-report, which may underestimate or overestimate the real figures [43]. 4) assessment of joint health was carried out using different clinical tools across studies, therefore it was not possible to performed meta-analyses. 5) almost all the studies in the meta-analyses were at high risk or unclear risk of bias in several domains, mainly in the blinding of participants and personnel, and blinding outcome assessment. 6) Overall, the main outcomes had a very low certainty of the evidence, mainly due to the risk of bias, inconsistency, and small sample size.

These limitations reflect the need for high-quality RCTs that compare different doses of prophylactic treatment (low, intermediate, or high doses) or extended half-life vs standard half-life CFCs. Which assess clinically relevant outcomes (mortality, ABR, ABJR, joint disease, joint status, pain, current health status-HRQol, activities, employment, educational attendance, resource utilization) [43], and with enough follow-up period to assess these outcomes (ideally more than one year).

However, this is an up-to-date summarize of the RCTs that have assessed different types of replacement therapies with CFCs in patients with hemophilia, which provides important information for decision-making in this regard.

## Conclusion

In conclusion, we found 9 RCTs, of which 6 compared episodic vs prophylactic treatment (at either low, intermediate or high doses), all of which were performed in patients with haemophilia A. ABR and AJBR were lower in the prophylaxis than in the episodic treatment. The results for ABR suggested a dose-response, while the results for AJBR did not. However, since the certainty of the evidence was very low for all the assessed outcomes, high-quality studies

that compare low, intermediate, and high prophylaxis doses are still needed to confirm these results and correctly inform the decision-making process regarding factor replacement therapies.

## Supporting information

**S1 Table. Search strategy.**
(PDF)

**S2 Table. Studies that were evaluated in full-text, and were excluded.**
(PDF)

**S3 Table. Additional results found in the studies.** *The risk in the intervention group (and its 95% confidence interval) is based on the assumed risk in the comparison group and the relative effect of the intervention (and its 95% CI). **CI**: Confidence interval; **eMRI**: Extended magnetic resonance imaging; **MD**: mean difference; **IQR**: Interquartile range; **SD**: Standard deviation; **HRQoL**: Health-related quality of life. **Explanations** a. We rated down one level for risk of bias. b. We rated down two levels for imprecision due to the small number of participants that presented the outcome (less than 200). c. We rated down one level for publication bias.
(PDF)

**S1 Checklist. PRISMA 2020 checklist.**
(DOCX)

## Author Contributions

**Conceptualization:** David García-Gomero, Stefany Salvador-Salvador, José Montes-Alvis, Celina Herrera-Cunti, Alvaro Taype-Rondan.

**Data curation:** Carolina J. Delgado-Flores, David García-Gomero, Alvaro Taype-Rondan.

**Formal analysis:** Carolina J. Delgado-Flores, Alvaro Taype-Rondan.

**Funding acquisition:** Carolina J. Delgado-Flores, David García-Gomero, Stefany Salvador-Salvador, José Montes-Alvis, Celina Herrera-Cunti.

**Investigation:** Carolina J. Delgado-Flores, David García-Gomero, Stefany Salvador-Salvador, José Montes-Alvis, Celina Herrera-Cunti.

**Methodology:** Carolina J. Delgado-Flores, José Montes-Alvis, Alvaro Taype-Rondan.

**Project administration:** Carolina J. Delgado-Flores, David García-Gomero, Stefany Salvador-Salvador, José Montes-Alvis, Celina Herrera-Cunti.

**Resources:** Carolina J. Delgado-Flores, David García-Gomero, Stefany Salvador-Salvador, José Montes-Alvis, Celina Herrera-Cunti.

**Software:** Carolina J. Delgado-Flores.

**Supervision:** Celina Herrera-Cunti, Alvaro Taype-Rondan.

**Validation:** David García-Gomero, Stefany Salvador-Salvador, José Montes-Alvis, Celina Herrera-Cunti, Alvaro Taype-Rondan.

**Visualization:** Carolina J. Delgado-Flores, Stefany Salvador-Salvador, José Montes-Alvis, Celina Herrera-Cunti.

**Writing – original draft:** Carolina J. Delgado-Flores, David García-Gomero, Stefany Salvador-Salvador, José Montes-Alvis, Celina Herrera-Cunti, Alvaro Taype-Rondan.

**Writing – review & editing:** Carolina J. Delgado-Flores, David García-Gomero, Stefany Salvador-Salvador, José Montes-Alvis, Celina Herrera-Cunti, Alvaro Taype-Rondan.

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
