## [Decision Letter · Decision Letter 0]

11 Aug 2021

PONE-D-21-14425

Effects of replacement therapies with clotting factors in patients with hemophilia: systematic review and meta-analysis

PLOS ONE

Dear Dr. Taype-Rondan, 

Thank you for submitting your manuscript to PLOS ONE. After careful consideration, we feel that it has merit but does not fully meet PLOS ONE’s publication criteria as it currently stands. Therefore, we invite you to submit a revised version of the manuscript that addresses the points raised during the review process.

We look forward to receiving your revised manuscript.

Kind regards,

Wolfgang Miesbach, MD

Academic Editor

PLOS ONE

1. Please ensure that your manuscript meets PLOS ONE's style requirements, including those for file naming. The PLOS ONE style templates can be found at https://journals.plos.org/plosone/s/file?id=wjVg/PLOSOne_formatting_sample_main_body.pdf and https://journals.plos.org/plosone/s/file?id=ba62/PLOSOne_formatting_sample_title_authors_affiliations.pdf.

2. Please include your tables as part of your main manuscript and remove the individual files. Please note that supplementary tables (should remain/ be uploaded) as separate "supporting information" files

3. We note that this manuscript is a systematic review or meta-analysis; our author guidelines therefore require that you use PRISMA guidance to help improve reporting quality of this type of study. Please upload copies of the completed PRISMA checklist as Supporting Information with a file name “PRISMA checklist”.

4. Thank you for submitting the above manuscript to PLOS ONE. During our internal evaluation of the manuscript, we found significant text overlap between your submission and the following previously published works, some of which you are an author.

- https://journals.plos.org/plosone/article?id=10.1371%2Fjournal.pone.0233220

Please revise the manuscript to rephrase the duplicated text, cite your sources, and provide details as to how the current manuscript advances on previous work. Please note that further consideration is dependent on the submission of a manuscript that addresses these concerns about the overlap in text with published work.

Reviewers' comments:

Reviewer's Responses to Questions

**Comments to the Author**

1. Is the manuscript technically sound, and do the data support the conclusions?

Reviewer #1: Yes

Reviewer #2: Yes

2. Has the statistical analysis been performed appropriately and rigorously? 

Reviewer #1: Yes

Reviewer #2: Yes

3. Have the authors made all data underlying the findings in their manuscript fully available?

Reviewer #1: Yes

Reviewer #2: Yes

4. Is the manuscript presented in an intelligible fashion and written in standard English?

Reviewer #1: Yes

Reviewer #2: Yes

5. Review Comments to the Author

Reviewer #1: This is an interesting and well-written paper

The conclusion is not unexpected as the benefits of prophylaxis are well established

This paper has the merit to provide an update on current status of knowledge

Maybe the authors should discuss and compare relative data on SHL and EHL-FVIII

In the title + abstract clearly indicate that only HA is covered

A table with a summary of what information should be collected in future trials could be useful.

Reviewer #2: This is an elegant review and meta-analysis by Taype-Rondan on the effect of clotting factor substitution in patients with haemophilia. The criteria of meta-analyses were met. It would still be interesting to see the authors' outlook on a possible future study design of trials that include the effect of long-acting factor preparation and nonfactor therapy.

6. PLOS authors have the option to publish the peer review history of their article (what does this mean?). If published, this will include your full peer review and any attached files.

Reviewer #1: No

Reviewer #2: No

---

## [Author Response · Author response to Decision Letter 0]

3 Nov 2021

Reviewer #1: 

• R1C1: This is an interesting and well-written paper. The conclusion is not unexpected as the benefits of prophylaxis are well established this paper has the merit to provide an update on current status of knowledge.

o We thank you for acknowledging the relevance of this work.

• R1C2: Maybe the authors should discuss and compare relative data on SHL and EHL-FVIII. 

o We agree, so we have included the following paragraph in the discussion (right before the “strengths and limitations” subheading:

“All studies found in our systematic review assessed interventions that used CFCs with standard half-life. CFCs with extended half-life require fewer CFCs administration per week due to their pharmacokinetic properties, which may improve the quality of life of the patients; and also require less use of medical devices, which could help lower the cost of prophylactic treatment. Thus, there is a need of RCTs assessing the use of extended half-life CFCs, which are believed to be as safe and effective as standard half-life CFCs, as WFH 2020 suggests. [6]”

• R1C3: In the title + abstract clearly indicate that only HA is covered. 

o We understand that our main results are based on hemophilia A, but this search aimed to find studies in patients with either hemophilia A or B. Although only one study on hemophilia B was found, and this has been described in the table 1 (characteristics of the studies).

• R1C4: A table with a summary of what information should be collected in future trials could be useful.

o We agree that this information is valuable, so we have included the following paragraph in the discussion (right before the “conclusion” subheading):

“These limitations reflect the need for high-quality RCTs that compare different doses of prophylactic treatment (low, intermediate, or high doses) or extended half-life vs standard half-life CFCs. Which assess clinically relevant outcomes (mortality, ABR, ABJR, joint disease, joint status, pain, current health status-HRQol, activities, employment, educational attendance, resource utilization) [43], and with enough follow-up period to assess these outcomes (ideally more than one year).”

Reviewer #2:

• R2C1: This is an elegant review and meta-analysis by Taype-Rondan on the effect of clotting factor substitution in patients with haemophilia. The criteria of meta-analyses were met.

o We thank you for your kind comment.

• R2C2: It would still be interesting to see the authors' outlook on a possible future study design of trials that include the effect of long-acting factor preparation and nonfactor therapy.

o We agree that this information is valuable, so we have included the following paragraph in the discussion (right before the “conclusion” subheading):

“These limitations reflect the need for high-quality RCTs that compare different doses of prophylactic treatment (low, intermediate, or high doses) or extended half-life vs standard half-life CFCs. Which assess clinically relevant outcomes (mortality, ABR, ABJR, joint disease, joint status, pain, current health status-HRQol, activities, employment, educational attendance, resource utilization) [43], and with enough follow-up period to assess these outcomes (ideally more than one year).”

---

## [Decision Letter · Decision Letter 1]

21 Dec 2021

Effects of replacement therapies with clotting factors in patients with hemophilia: a systematic review and meta-analysis

PONE-D-21-14425R1

Dear Dr. Taype-Rondan,

We’re pleased to inform you that your manuscript has been judged scientifically suitable for publication and will be formally accepted for publication once it meets all outstanding technical requirements.

Kind regards,

Wolfgang Miesbach, MD

Academic Editor

PLOS ONE

Reviewers' comments:

Reviewer's Responses to Questions

**Comments to the Author**

1. If the authors have adequately addressed your comments raised in a previous round of review and you feel that this manuscript is now acceptable for publication, you may indicate that here to bypass the “Comments to the Author” section, enter your conflict of interest statement in the “Confidential to Editor” section, and submit your "Accept" recommendation.

Reviewer #2: All comments have been addressed

2. Is the manuscript technically sound, and do the data support the conclusions?

Reviewer #2: Yes

3. Has the statistical analysis been performed appropriately and rigorously? 

Reviewer #2: Yes

4. Have the authors made all data underlying the findings in their manuscript fully available?

Reviewer #2: Yes

5. Is the manuscript presented in an intelligible fashion and written in standard English?

Reviewer #2: Yes

6. Review Comments to the Author

Reviewer #2: (No Response)

7. PLOS authors have the option to publish the peer review history of their article (what does this mean?). If published, this will include your full peer review and any attached files.

Reviewer #2: No

---

## [Editor Report · Acceptance letter]

27 Dec 2021

PONE-D-21-14425R1 

Effects of replacement therapies with clotting factors in patients with hemophilia: a systematic review and meta-analysis 

Dear Dr. Taype-Rondan:

I'm pleased to inform you that your manuscript has been deemed suitable for publication in PLOS ONE. Congratulations! Your manuscript is now with our production department. 

Kind regards, 

on behalf of

Dr. Wolfgang Miesbach 

Academic Editor

PLOS ONE